# Role of Angiotensin II in Cardiovascular Diseases: Introducing Bisartans as a Novel Therapy for Coronavirus 2019

**DOI:** 10.3390/biom13050787

**Published:** 2023-05-02

**Authors:** Jordan Swiderski, Laura Kate Gadanec, Vasso Apostolopoulos, Graham J. Moore, Konstantinos Kelaidonis, John M. Matsoukas, Anthony Zulli

**Affiliations:** 1Institute for Health and Sport, Victoria University, Melbourne, VIC 3030, Australia; jordan.swiderski@live.vu.edu.au (J.S.); laura.gadanec@live.vu.edu.au (L.K.G.); vasso.apostolopoulos@vu.edu.au (V.A.); 2Immunology Program, Australian Institute for Musculoskeletal Science, Melbourne, VIC 3021, Australia; 3Pepmetics Incorporated, 772 Murphy Place, Victoria, BC V8Y 3H4, Canada; mooregj@shaw.ca; 4Department of Physiology and Pharmacology, Cumming School of Medicine, University of Calgary, Calgary, AB T2N 4N1, Canada; 5NewDrug PC, Patras Science Park, 26500 Patras, Greece; k.kelaidonis@gmail.com; 6Department of Chemistry, University of Patras, 26504 Patras, Greece

**Keywords:** angiotensin-converting enzyme 2, angiotensin II, bisartans, cardiovascular diseases, coronavirus 2019, severe acute respiratory syndrome coronavirus 2, renin–angiotensin system

## Abstract

Cardiovascular diseases (CVDs) are the main contributors to global morbidity and mortality. Major pathogenic phenotypes of CVDs include the development of endothelial dysfunction, oxidative stress, and hyper-inflammatory responses. These phenotypes have been found to overlap with the pathophysiological complications of coronavirus disease 2019 (COVID-19). CVDs have been identified as major risk factors for severe and fatal COVID-19 states. The renin–angiotensin system (RAS) is an important regulatory system in cardiovascular homeostasis. However, its dysregulation is observed in CVDs, where upregulation of angiotensin type 1 receptor (AT_1_R) signaling via angiotensin II (AngII) leads to the AngII-dependent pathogenic development of CVDs. Additionally, the interaction between the spike protein of severe acute respiratory syndrome coronavirus 2 with angiotensin-converting enzyme 2 leads to the downregulation of the latter, resulting in the dysregulation of the RAS. This dysregulation favors AngII/AT_1_R toxic signaling pathways, providing a mechanical link between cardiovascular pathology and COVID-19. Therefore, inhibiting AngII/AT_1_R signaling through angiotensin receptor blockers (ARBs) has been indicated as a promising therapeutic approach to the treatment of COVID-19. Herein, we review the role of AngII in CVDs and its upregulation in COVID-19. We also provide a future direction for the potential implication of a novel class of ARBs called bisartans, which are speculated to contain multifunctional targeting towards COVID-19.

## 1. Introduction

Despite recent advances in understanding the mechanisms and potential therapeutic interventions, cardiovascular diseases (CVDs) remain the leading cause of global morbidity and mortality [1]. Globally, CVDs account for an estimated 19.1 million deaths, equating to approximately 30% of all deaths in a given year [2]. CVD is an umbrella term that encompasses any pathology that affects the heart and vasculature, such as hypertension, coronary heart disease, heart failure, atherosclerosis, acute myocardial injury, myocarditis, and stroke [3]. Hypertension, clinically defined as abnormally elevated blood pressure, is the most prevalent CVD and affects an estimated 874 million people [2]. Moreover, hypertension was considered the highest risk factor for premature cardiovascular-related deaths in 2021 [2]. Furthermore, it is estimated that 244.1 million people are living with some form of heart disease [2]. Due to an aging population, an increase in sedentary lifestyles, and a rise in obesity, current projections estimate that the prevalence of CVDs is expected to grow by 45% by 2035 [4]. The pathogenesis of CVDs is extremely complex and involves overlapping and multifaceted crosstalk between signaling mechanisms. This contributes to the pathophysiological development of the major phenotypes that are identified in the progression of various CVDs, including endothelial dysfunction, oxidative stress, and a hyper-inflammatory response [5]. One of the major signaling pathways is the renin–angiotensin system (RAS), a crucial hormonal regulatory system responsible for maintaining cardiovascular, renal, pulmonary, immune, and neural homeostasis [6]. It has become apparent that dysregulation of RAS is present in the pathophysiology of CVDs, in which the upregulation of toxic vasoactive peptides can function to mediate tissue damage and eventual multi-organ failure [7,8,9]. 

Since its emergence at the end of 2019, coronavirus disease 2019 (COVID-19), caused by the severe acute respiratory syndrome coronavirus-2 (SARS-CoV-2), has contributed to over 757 million cases and 6.9 million deaths [10]. Initially, COVID-19 was identified as a respiratory disease; however, pathological cardiovascular conditions have a marked prevalence in infected individuals [11,12]. Large cohort studies have shown that this increased prevalence of CVDs is associated with the progression of the disease into severe and fatal states [7,13]. Given the magnitude of viral exposure, this correlation is of significant concern due to the increasing frequency of CVDs in the global population. 

The hallmark of COVID-19 pathogenesis is oxidative cellular damage, endothelial dysfunction, and an unregulated hyper-inflammatory response, which has been linked to the progression of systemic cytokine production, or “cytokine storm” [7,14,15,16]. These pathophysiological features are also present in CVDs, suggesting similar mechanisms of disease [7,14,15,16]. Additionally, the interaction between SARS-CoV-2 and angiotensin-converting enzyme-2 (ACE2), an important regulating protein of the RAS, for viral entry appears to be evident in establishing a relationship between CVDs and COVID-19 [17]. 

Herein, we focus on the role of angiotensin II (AngII), a vasopressor peptide of the RAS, in CVDs and examine how its upregulation might contribute to the underlying pathology of COVID-19, especially in individuals who have one or more cardiovascular comorbidities and are at a greater risk of developing severe disease. We also discuss the efficacy of angiotensin type 1 receptor (AT_1_R) blockers in the treatment of COVID-19 and provide a hypothesis for the future directions of cardiovascular and COVID-19 management. 

## 2. The Role of AngII in CVDs

AngII is a major vasoactive octapeptide of the RAS, and almost all its known effects are carried out through the activation of the G-protein-coupled receptor AT_1_R [6]. The AngII/AT_1_R interaction is important for maintaining vascular tone through vasoconstrictive measures, as well as increasing aldosterone release for water and electrolyte retention, promoting cellular growth, activating immune cells, and mediating pro-oxidative, pro-fibrotic, and pro-thrombotic effects [18,19]. The synthesis of AngII occurs through the classical axis of the RAS, in which angiotensinogen produced by the liver is converted into the inactive decapeptide angiotensin I (AngI) by the release of renin into circulation via the kidneys [20]. AngI is further hydrolyzed and converted into the main effector protein AngII by the angiotensin-converting enzyme (ACE), which mediates its tissue-dependent effects through the AT_1_R [21]. Under normal conditions, the synthesis and signaling of AngII are regulated through the counter-regulatory axis of the RAS [22]. This pathway primarily consists of the ACE2, angiotensin (1–7) (Ang (1–7)), and the Mas1 oncogene receptor (MasR) [22]. Ang (1–7) promotes synergistic effects to AngII, resulting in vasodilative, anti-inflammatory, anti-proliferative, anti-fibrotic, and anti-thrombotic properties via MasR signaling. ACE2 degrades AngII to Ang (1–7), thereby reducing the effects of AngII and AT_1_R activation and increasing the cardio-protective effects of Ang (1–7) [23]. AngII is also metabolized by aspartate decarboxylase to produce angiotensin A (AngA), which can be converted into the recently discovered peptide almandine (ALM) by ACE2 [24]. ALM has been observed to exhibit protective AngII antagonistic effects similar to Ang (1–7) through activation of the Mas-related G-protein-coupled receptor D (MrgDR) signaling pathway [25]. Our group has previously highlighted the cardioprotective role of ALM in mitigating endothelial dysfunction through MrgDR activation [26]. ALM has also been shown to provide anti-inflammatory [27,28], anti-oxidative [29,30], and pulmonary protective effects [31]. Moreover, AngII can be further processed by aminopeptidase A to form angiotensin (2–8) (Ang III) [32]. AngIII has a similar affinity to the AT_1_R as AngII [33]. Its major physiological role is in the brain, where it acts as a central regulator of blood pressure and vasopressin release [34]. AngIII can be cleaved by alanyl aminopeptidase N to generate angiotensin IV (AngIV). This peptide has been shown to bind to the angiotensin type 4 receptor (aka IRAP), resulting in the mitigation of cardiac hypertrophy [35], fibrosis [35], and inflammation [36], as well as inducing vasodilation through nitric oxide release [37,38]. Interestingly, there is evidence that AngIV may act as a weak agonist of the AT_1_R, resulting in vasoconstriction [39]. The cardioprotective roles of ALM and AngIV remain largely elusive, and further investigation is required to understand their physiological roles in the cardiovascular system. While the two axes are maintained at equilibrium, an increase in AngII in the circulation can result in the overstimulation of the AT_1_R, leading to the unregulated production of toxic effects such as oxidative stress, secretion of pro-inflammatory mediators to produce a hyper-inflammatory response, and cytokine storm, as well as fibrosis and damage to vascular smooth muscle cells (VSMCs), ultimately contributing to the pathogenesis of CVDs [40]. 

The AT_1_R signaling cascade is a highly complex pathway that cross-talks with various other intracellular mechanisms to mediate numerous tissue-dependent physiological effects [41]. In the regulation of blood pressure, when AngII activates the AT_1_R, it induces confirmational changes in the adjacent G-proteins, leading to the release of phospholipid-C (PLC), which in turn hydrolyzes membrane lipids to induce several downstream effects. These include the liberation of diacylglycerol, which can activate protein kinase C (PCK), and inositol-1,4,5-triphosphate (IP3) [42]. IP3 can then bind to its receptor on the sarcoplasmic reticulum to immobilize the release of Ca^2+^ into the cytoplasm [43]. Ca^2+^ is then free to activate the phosphorylation of myosin to enhance its interaction with actin to induce VSMC contraction [41,44,45]. Additionally, AT_1_R can also activate various downstream protein kinases, such as mitogen-activated protein kinases (MAPK), extracellular signal-regulated kinases (ERK), and Janus kinases-STAT (JAK-STAT) signaling pathways, that are involved in the cardiovascular pathologic effects of AngII [46,47] (Figure 1). 

### 2.1. AngII and Endothelial Dysfunction

Endothelial dysfunction is considered the earliest vasculature abnormality in hypertension, atherosclerosis, and heart disease [48,49]. It is characterized by diminished vasorelaxation properties as a result of a loss in vascular integrity, inflammation, and can cause unregulated VSMC growth [50]. Chronic stimulation of the AT_1_R is associated with hypertension and can result in prolonged vasoconstriction, which leads to excessive mechanical stress on the microvasculature. This stress can trigger VSMCs to undergo adaptational changes, including remodeling, hypertrophy, hyperplasia, and fibrosis [51,52]. These changes result in decreased arterial compliance, which is associated with endothelial dysfunction [51,52]. In addition, AngII/AT_1_R signaling is known to have a direct effect on the promotion of endothelial dysfunction through AngII-dependent activation of PKC, MAPK, ERK1/2, and p38-MAPK signaling pathways in VSMCs [51,53] (Figure 1). AngII has been shown to increase p38-MAPK, resulting in reduced nitric oxide (NO) via uncoupling of endothelial nitric oxide synthase (eNOS) [54]. NO is responsible for maintaining vascular tone as well as preventing unregulated cell growth and oxidative stress in VSMCs. Thus, its downregulation is crucial in the development of endothelial dysfunction [55].

AngII promotes pro-fibrotic growth factors involved in the progression of cardiac fibrosis, as seen by the increase in mRNA expression of tissue growth factor-β (TGF-β) and nuclear factor kappa B (NF-κB) [56]. AngII promotes excessive accumulation of collagen in vascular tissue, leading to cardiac dysfunction as well as cardiac remodeling [57]. Mechanisms that may be implicated in this association include endothelial dysfunction, hypertrophy, fibrosis, and cellular apoptosis [58]. 

### 2.2. AngII and Oxidative Stress

Oxidative stress is a result of an imbalance between antioxidant and free radical production, with increased, reactive oxygen species (ROS) that can induce cellular damage and apoptosis, resulting in tissue damage [59]. Oxidative stress is closely associated with endothelial dysfunction and the promotion of vascular inflammatory responses in various CVDs [59]. AngII has been observed to be the main contributor to the pathogenic effects of oxidative stress in CVDs [60]. AngII/AT_1_R-dependent signaling in VSMCs has been shown to result in the activation of nicotinamide adenine dinucleotide phosphate oxidase (NOX) activity through upstream regulation of MAPK, PKC, and NF-κB pathways, promoting the production of superoxide ions and hydrogen peroxide [60] (Figure 1). In endothelial dysfunction, superoxide ions can react with NO to produce peroxynitrite [60], reducing the bioavailability of NO and preventing its ability to induce vasodilation and regulate vascular tone [60]. AngII-induced hypertension in rats was demonstrated to increase NOX activity, leading to the production of superoxide ions and impairing vascular relaxation by decreasing NO [61]. Excessive production of ROS can result in mitochondrial dysfunction, DNA damage, and damage to other cellular components that further impair vascular structure and function [62]. ROS produced by NOX units is closely related to cardiomyocyte cell proliferation and differentiation as a result of ROS-induced upregulation of MAPK and other kinase-dependent signaling pathways to promote cardiac hypertrophy, fibrosis, and impaired vascular tone [63]. Additionally, ROS production by AngII-mediated NOX activation is known to be directly involved in the promotion of endothelial growth factors to contribute to VSMC remodeling, hypertrophy, and inflammation through subsequent activation of ERK1/2, NF-κB and additional kinase-dependent signaling pathways [64,65,66]. ROS damage to DNA can also promote inflammation through NF-κB to increase the production of pro-inflammatory cytokines [67] (Figure 1). 

### 2.3. AngII and CVD Inflammation

Inflammation plays a pivotal role in the pathology of CVDs. Several pro-inflammatory mediators are upregulated in both human and animal models [68,69]. Heart disease, atherosclerosis, and hypertension are all associated with increased pro-inflammatory cytokines, such as tumor necrosis factor-alpha (TNF-α), interleukin (IL)-1β, IL-6, and interferon-gamma (IFN-γ) [70] (Figure 1). These cytokines significantly contribute to the pathways of oxidative stress, cardiac hypertrophy, fibrosis, coagulation, hyper-inflammation, and endothelial dysfunction [71].

In addition, RAS has a key role in regulating inflammation. AngII/AT_1_R signaling is known to mediate the activation of NF-κB [72], an important transcription factor that is responsible for the upregulation of cell adhesion molecules, proliferation/cellular growth factors, and the production of proinflammatory cytokines and chemokines [73]. AT_1_R-activated MAPK/NF-κB signaling pathways have also been observed to trigger the amplification of IL-6, which is an important immunoinflammatory cytokine responsible for fever and acute phase inflammation. AngII infusions are observed to directly enhance the production of serum cytokines TNF-α, IL-6, IL-1β, and IFN-γ in aortic tissue [74]. Additionally, isolated peripheral blood T-cells produce a greater concentration of TNF-α and IFN-γ in response to AngII infusion in mice compared to the control [75]. 

AngII has been shown to directly stimulate the secretion of adhesion molecules such as vascular cell adhesion molecule-1, selectins (P- and L-selectin), and potent monocyte chemoattractant protein-1 (MCP-1). These molecules promote the migration and accumulation of immune cells in local tissue and contribute to the inflammatory response in CVDs [76]. Infiltration of immune cells is further associated with the secretion of transforming growth factor beta 1 (TGF-1β) and IL-1β, which is promoted by chronic activation of AngII [77,78]. These cytokines contribute to the conditions developing in heart failures, such as vascular fibrosis and stiffening [79]. Ang II-induced fibrosis is associated with an altered expression of TGF-β, TNF-α, MCP-1, and IL-6 [80]. One study observed that in vitro infusion of AngII in mouse models resulted in dose-dependent progression of endothelial dysfunction, which was associated with an increase in IL-6 and macrophage count [81]. Additionally, an AngII-dependent increase in serum IL-6 concentration is also shown to induce downstream activation of STAT3 pathways and inhibit eNOS expression, leading to a decrease in NO and the promotion of endothelial dysfunction [82]. IFN-γ can also regulate the JAK-STAT pathway to increase angiotensinogen expression, which can then result in the perpetual upregulation of angiotensinogen conversion to AngII through the RAS pathway [83]. 

## 3. The Relationship between CVDs and COVID-19

There is overwhelming evidence that CVDs are risk factors for the progression of COVID-19 [84]. It is believed that 75% of all severe cases contain at least one cardiovascular condition [85], while other studies have shown that the development of heart failure and myocardial damage is increased during the critical stages of COVID-19 progression [86]. Autopsy reports from deceased COVID-19 patients highlighted the prevalence of cardiac pathologies, with an increased frequency of myocardial inflammation, thrombosis, and elevated cardiac injury biomarkers present [13,87]. A retrospective study of over 12,000 hospitalized COVID-19 patients reported that cardiovascular comorbidities, such as coronary heart disease, and heart failure, were significantly more common in non-survivors. Predisposition to hypertension was also observed to have a higher independent prognostic value for mortality [88]. 

### COVID-19 and RAS

The respiratory tract and lungs have been established as the primary sites for the entry of SARS-CoV-2 [17]. This is established through the interaction of the viral spike protein (S-protein) with the extracellular receptor binding domain (RBD) of ACE2 on airway epithelial type II alveolar cells [17]. This interaction has been identified to result in the downregulation of cellular ACE2 expression via (1) internalization of the ACE2-viral complex and/or (2) shedding of the extracellular domain of ACE2 caused by the presence of host and viral proteases [89,90]. This downregulation of ACE2 expression and loss of its functional ability to metabolize AngII into the protective peptide Ang (1–7) results in a shift in the RAS to favor AngII/AT_1_R signaling. This may contribute to the pathogenic cardiovascular complications that arise in COVID-19, such as vascular dysfunction, oxidative stress, and a proinflammatory “cytokine storm” [23] (Figure 1). In patients with acute respiratory distress syndrome (ARDS) admitted to the intensive care unit, the plasma ratio of Ang (1–7) was observed to be higher in survivors than non-survivors. This suggests that greater ACE2 expression may be a protective factor against severe lung injury due to the ability of ACE2 to degrade AngII into the protective peptide Ang (1–7) [91]. In mice with heart failure due to myocardial infarction, deficiency of ACE2 with an increase in myocardial infarct size, along with an increase in ERK1/2/JNK molecular signaling pathways. This led to an increase in the expression of IL-6, MCP-1, and MMP-9. Furthermore, overexpression of ACE2, Ang (1–7), and ALM receptor MrgD has been shown to improve cardiac remodeling during MI [92,93,94,95]. Several case reports, along with clinical phase I and II studies, have reported that recombinant ACE2 may improve the clinical course of patients with COVID-19 by increasing the degradation of AngII into Ang (1–7), as well as increasing ALM [96,97,98]. Therefore, ACE2 downregulates the toxic pathophysiological effects of AngII while upregulating the protective counter-regulator axis.

Severe and fatal forms of COVID-19 have been associated with a potent inflammatory response known as a “cytokine storm,” which shares underlying signaling mechanisms with hypertension and other CVDs [99]. Inflammation of the heart myocardium (myocarditis) is recognized as a high-risk mortality factor for heart failure in COVID-19 due to the augmented increase in inflammatory mediators that occurs in the cytokine storm [11]. The cytokine storm is identified by the excessive increase in proinflammatory mediators such as IL-6, Il-1, IL-1β, IL-8, MCP-1, matrix metalloproteins-1 and -3, and TNF-α [100] (Figure 1). NF-κB has been identified as a key mediator in the pathogenesis of the cytokine storm [101]. The immune response during SARS-CoV-2 infection is shown to be mediated by macrophage-dependent activation of pattern recognition receptors, such as toll-like receptor 4 (TLR4), resulting in exacerbated immune and pro-inflammatory responses through NF-κB [102,103]. These pathways are also induced by dysregulation of the RAS. The AT_1_R is expressed on the surface of various immune cells, including T-lymphocytes (T-cells), which can promote the proliferation, migration, adhesion, and differentiation of immune cells to contribute to the immune response [104,105]. AngII has been shown to promote the infiltration of immune cells into myocardial tissue as well as regulate macrophage polarization via TLR4/NF-κB/MAPK signaling to induce oxidative stress, mitochondrial dysfunction, and apoptosis [106,107]. Additionally, AngII/AT_1_R activation of NF-κB/STAT can mediate the amplification of IL-6 via a positive feedback loop by which IL-6 activation of the IL-6 receptor can activate NF-κB /STAT3 to further release IL-6 [15]. This IL-6 amplifying mechanism can induce chronic inflammation and further increase the production of proinflammatory cytokines to create a cytokine storm and increase the severity of SARS-CoV-2 infection [108]. IL-6 is predominantly relevant as its elevated plasma levels in COVID-19 patients are directly related to the severity of the disease [109]. Although evidence has suggested that an increase in T-cell activation and proliferation may be beneficial for viral clearance [110] and for improving disease state [111], excessive activation, specifically of the Th17 phenotype, may contribute to the production of pro-inflammatory mediator and inhibit the regulatory function of protective T-cells. This can contribute to the progression of tissue damage in COVID-19 [112]. A recent study observed that AngII stimulation of human mononuclear cells resulted in a significant increase in CD4+ Th1/Th17 and CD8+ Th17 phenotype cells along with an increase in proinflammatory cytokines TNF-α, IFN-γ, and IL-17, thereby exacerbating the inflammatory profile [113]. A positive correlation has been established between T-cell and IL-17 production and COVID-19 severity [114]. Furthermore, T-cells are shown to be deregulated in hypertension and CVDs, resulting in an increased production of the pro-inflammatory cytokines IL-6, IL-7, IFN-γ, and TNF-α [115,116]. AngII-dependent activation of p38-MAPK contributes to endothelial dysfunction and end-organ damage in CVDS [117], and it is also seen to play a role in T-cell activation [118]. These immunoinflammatory mechanisms of AngII and CVDs overlap with the characteristics of the “cytokine storm” that develops in severe COVID-19, and thus, in conjunction with one another, may contribute to the accelerated end-organ damage.

Due to the role of AngII/AT_1_R signaling in vascular remodeling and fibroblast activity, AngII is also thought to be implicated in lung injury and the pathogenesis of pulmonary fibrosis [119]. RAS dysregulation in the lungs is associated with airway inflammation, fibrosis, and pulmonary hypertension [120]. Collectively, these observations in the overlap between COVID-19 and AngII/AT_1_R signaling in CVD pathology indicate that when combined, RAS dysregulation may contribute to an exacerbated inflammatory response and a greater degree of tissue damage to increase the severity of disease progression.

The association between SARS-CoV-2 infection, RAS dysfunction, and a potential increase in AngII is evident in several experimental and clinical studies. Murine models of SARS-CoV, which share a 79.5% sequence identity with SARS-CoV-2 [121], have established molecular links between SARS-CoV pathogenesis and RAS dysregulation. Injection of the SARS-COV-2 S-protein into wild-type mice resulted in a decreased expression of ACE2 in the lungs, which was associated with significant increases in AngII, respiratory inflammation, vascular permeability, and decreased oxygen saturation [122]. These results were associated with worsening respiratory inflammation and an increase in vascular permeability, which was attenuated by both downregulations of AT_1_R expression and through the administration of AT_1_R antagonists [122,123]. A study of Huh7.5 and A549 cell lines infused with the SARS-CoV-2 S-protein demonstrated that cellular infection was associated with a significant decrease in ACE2 expression and a significant increase in AT_1_R expression, which was associated with upregulated MAPK/NF-κB signaling and an increase in IL-6 [124]. Elevated levels of IL-6 have been identified as a major factor in the exacerbation of the COVID-19 inflammatory response, contributing to vascular inflammation, acute respiratory distress syndrome development, and mechanical ventilation [125]. Overstimulation of AngII/AT_1_R-induced signaling is observed to promote endothelial dysfunction, contributing to the development of ARDS as characterized by the onset of severe hypoxia, inflammatory cell accumulation, and pulmonary oedema [11]. Clinical studies have indicated, through univariate analysis, a positive correlation between AngII levels and COVID-19 severity [126]. A reduction in ACE2 expression and an increase in both AngI and AngII have also been observed in patients presenting with COVID-19 compared to healthy controls [127]. The severity of COVID-19 in hospitalized patients has been observed to be associated with a marked increase in serum AngII concentration. This increase was also positively correlated with a greater viral load and lung injury through ARDS. These findings suggest that viral infection may increase AngII via AT_1_R to result in organ-damaging effects and contribute to a worsening disease outcome [128]. Furthermore, another study that evaluated the levels of the RAS proteins and peptides in patients with COVID-19 observed that although ACE, ACE2, and Ang (1–7) levels were similar in patients with critical and severe COVID-19, AngII was significantly higher in patients in critical conditions [129]. This indicates a possible impairment in AngII conversion. It was concluded that early measurement of AngII could be a potential clinical biomarker for identifying patients at higher risk for severe disease progression. Interestingly, a SARS-CoV-2 S-protein mediated cell fusion assay in human epithelial Calu-3 cells showed that AngII acted on the AT_1_R to increase ACE2 expression, which enhanced SARS-CoV-2 infection [130]. These effects were abolished by ARBs [130]. Therefore, enhanced production of AngII in patients with a dysregulated RAS (such as those with cardiovascular comorbidities) could put them at a greater risk for COVID-19 infection.

Evidence highlights the importance of vascular health in COVID-19 disease progression due to the exacerbation of hyper-inflammation, immune cell activation, and oxidative stress, resulting in the formation of a damaging cytokine storm along with vascular dysfunction to cause a greater degree of end-organ damage. Clinical evidence has shown that severe COVID-19 can develop as a result of altered vascular function, leading to inflammation and the development of CVDs [131,132]. Thus, excess AngII, unopposed by adequate Ang (1–7) and accompanied by the loss of ACE2, may contribute to the increase in AT_1_R signaling responsible for vascular dysfunction and the progression of cardiovascular complications.

## 4. ARBs as a Treatment for COVID-19

Angiotensin receptor blockers (ARBs) are a class of well-tolerated pharmaceuticals currently prescribed to patients with hypertension, cardiovascular disease, and renal disease [133]. ARBs are known to have a low incidence of side effects and contain an excellent tolerability profile comparable to placebo [134,135]. Randomized control trials have demonstrated that the most common adverse effects of ARB treatment include headaches, dizziness, and fatigue [136,137,138]. In rare cases, the major adverse effects of ARBs are hemodynamic, resulting in an increased risk of hypotension [139,140]. Additionally, in patients with renal disease, ARBs may increase the risk of hyperkalemia due to the direct effect of AT_1_R inhibition on renal electrolyte balance [141]. It is important to recognize that these effects are primarily pronounced in patients taking a combination of ARBs and other blood pressure-lowering medication.

Due to the potential relationship between CVDs, COVID-19, and RAS dysregulation, growing interest has mounted in the potential therapeutic benefits of ARBS in COVID-19 management [142,143]. Modulation of the AT_1_R by ARBs has shown attenuation of the release of proinflammatory cytokines TNF-α, and IL-6, as well as a decrease in ROS production through downregulation of MAPK/NF-κB pathways [144,145,146]. In mice, ARBs are effective in suppressing T-cell proliferation and inhibiting IFN-γ production. Experimental evidence has shown that ARBs are effective in reducing AngII-mediated endothelial dysfunction, improving vascular dysfunction, and preventing respiratory damage [147]. This leads to the speculation that ARBs may be an effective therapeutic tool in the prevention of severe COVID-19 [147]. Several clinical studies have demonstrated an association between ARBs and improved outcomes in hypertensive COVID-19 patients [148,149,150,151,152] (Table 1). The use of ARBs has been seen to significantly decrease disease severity during hospitalization compared to non-ARB RAS inhibitors [153]. A meta-analysis of thirty studies evaluating the safety and efficacy of ARBs in patients with COVID-19 concluded that the administration of ARBs was associated with a significant reduction in the risk of severe and fatal outcomes [154]. ARBs, candesartan and telmisartan, have been identified through comparative analysis to be the most promising candidates for COVID-19 management [155,156], with recent multicenter clinical trials indicating that these drugs are effective in reducing the morbidity and mortality of hospitalized COVID-19 patients due to their anti-inflammatory and vascular protective effects [157,158]. Telmisartan is known to contain a slow dissociation from AT_1_R that results in an apparently irreversible blockage of A that could be very beneficial in preventing an increase in circulating AngII in COVID-19 [159,160]. Crucially, a recent study assessing the therapeutic efficacy of ARB candesartan in SARS-CoV-2 infected cell lines observed through transcriptome analysis that viral infection was positively associated with the upregulation of 210 genes involved in pro-inflammatory cytokine-mediated signaling pathways, ROS production, immune cell activation, and pro-fibrotic and pro-coagulative mediation [161]. Administration of candesartan was observed to normalize these genes, strongly suggesting that this ARB could be beneficial in the treatment of COVID-19 [161]. It is important to note that current ARB clinical studies include a variety of ARBs with a common limitation of not having enough Power to detect differences between individual drugs [162]. 

While this evidence supports the use of ARBs in COVID-19 treatment, a major concern among clinical physicians is the potential ability of RAS inhibitors to upregulate ACE2 expression and potentially affect the welfare of at-risk patients currently receiving the medication. This intuition is of concern, as an increase in ACE2 expression in the lungs may promote a greater infection risk of cells with SARS-CoV-2. Current evidence on increased expression of ACE2 with RAS blockers is based on renal studies [167] and limited experimental studies [168,169]. An important study exploring the in vitro expression of ACE2 and AT_1_R in response to ARB treatment in Vero E6 cells showed that ARBs increased the expression of ACE2 and AT_1_R [170]. Treatment with the ARB irbesartan was also associated with an increase in viral RNA, demonstrating the potential ability of ARBs to increase susceptibility to viral infection [170]. It is important to note that this study was carried out independently of AngII and that the Vero E6 cell line for SARS-CoV-2 replication remains to be validated [171]. Additionally, the authors report that these results were not supported in both human renal and intestinal cell lines [171]. Moreover, agents that upregulate ACE2 have gained notoriety as potential COVID-19 treatments, as they may reduce the pathogenic effect of AngII, reactivate ACE2, and ultimately restore RAS balance [142,172,173]. For example, our group has recently shown that the putative ACE2 activator, diminazene aceturate, can interact with the SARS-CoV-2 S-protein and reduce AngII-mediated constriction in rabbit iliac arteries [172,173]. To date, there is no conclusive evidence regarding the increased risk of infectivity, morbidity, or mortality among RAS inhibitor users. Moreover, clinical studies have failed to identify a link between ARBs and increased infectivity [174]. Substantial clinical studies further demonstrate that the use of ARBs does not worsen clinical presentations and is seen to significantly reduce mortality and morbidity, especially in patients presenting with one or more cardiovascular comorbidities [133,149,150,151,155].

## 5. Bisartans, a New Generation of ARBs against SARS-CoV-2

Based on the evidence presented in this review, it is understood that pre-existing CVDs may exacerbate COVID-19 severity via an AngII-dependent mechanism. Therefore, targeting these mechanisms may be a potential therapeutic target for diminishing COVID-19 severity. Here, we propose a novel class of ARBs, termed “bisartans,” that we identified to display multifunctional properties that may be beneficial for the management of hypertension, CVDs, and COVID-19 [175]. Preliminary studies conducted by our group have shown the ability of bisartans to significantly antagonize the AT_1_R through in silico molecular modeling as well as ex vivo rabbit vascular studies [175,176]. Although ARBs possess common structural features required for effective antagonism of AT_1_R, molecular modeling studies suggest that unique minor differences in the chemical structure of ARBs may relate to differences in their characteristic binding kinetics and pharmacological bioactivity. These differences can contribute to their effectiveness against cardiovascular etiologies [160,177]. The strength of receptor antagonistic behavior is correlated to the number of salt bridges formed between the ARB and specific residuals of the AT_1_R, namely Arg^167^ and Tyr^35^ [178,179]. In contrast to other ARBs, bisartans are synthesized to contain a bis-alkylated imidazole core with two symmetrically anionic tetrazole groups to provide additional salt-bridge interactions with AT_1_R residuals Arg^167^ and Lys^199^ to induce overwhelming, insurmountable antagonism that cannot be overcome by increases in AngII concentration (Figure 2). Previous in silico modeling has identified those bis-substituted imidazole analogs that contain biphenyl moieties (such as the biphenyl tetrazole groups present in bisartan) as having a greater antagonistic activity towards the AT_1_R in comparison to the carboxyl and tetrazole groups of commercial ARBs [180]. Additionally, bisartans are designed to mimic a charge relay system (CRS) present within the interaction between AngII and AT_1_R, with the N-carboxylate imidazole of bisartan located at the correct distance to provide an additional salt-bridge interaction with the Tyr^35^ residual, potentially increasing the strength of binding affinity significantly further [175,176] (Figure 2). Interestingly, dynamic molecular simulations conducted by our group on bisartans revealed that they possess effective docking to the Zn^2+^ domain of the ACE2/S-protein RBD complex [175] (Figure 2). Additionally, bisartans were identified to be more effective than Nirmatrelvir, the active compound of Pfizer’s antiviral COVID-19 drug, in disrupting the CRS of 3-chymotrypsin-like protease (3Cl-pro), a protease required for SARS-CoV-2 replication, which we have identified as a key target for drug development against viral infection [12,175]. The ionic tetrazole groups of bisartans were found to have a more stable interaction with the Cys^145^ residual of 3Cl-pro than the nitrile pharmacophoric group of nirmatrelvir, resulting in greater inhibition of the CRS protease mechanism [175] (Figure 2). As a result, bisartans may be more effective than nirmatrelvir in preventing the synthesis of essential SARS-COV-2 proteins required for viral replication in humans. These preliminary investigations suggest that the properties of bisartans may contain greater efficacy and potency than current pharmaceuticals for the treatment of RAS dysregulation in hypertension, CVDs, and COVID-19, as well as potentially being a more suitable multifunctional drug with antiviral drug characteristics to also prevent SARS-CoV-2 infection. 

## 6. Conclusions

This literature review highlights the profound dysregulation of the RAS in the pathogenesis of COVID-19. Current evidence suggests that CVDs may exacerbate COVID-19 severity via AngII-dependent mechanisms and viral targeting of ACE2. This can result in dysfunction and reduced expression of the protective counter-regulatory axis (ACE2/Ang (1–7)/MasR) and overexpression of the deleterious classical axis (AngII/AT_1_R) of the RAS. This is characterized by vasoconstriction, vascular remodeling, oxidative stress, and hyper-inflammation. Based on this association, ARBs may be an effective treatment for COVID-19. Additionally, we present “bisartans,” a novel class of AT_1_R inhibitors that exhibit multifaceted antiviral abilities against SARS-CoV-2 infection. Therefore, bisartans may be a promising treatment for hypertension, CVDs, and COVID-19.

## Figures and Tables

**Figure 1 biomolecules-13-00787-f001:**
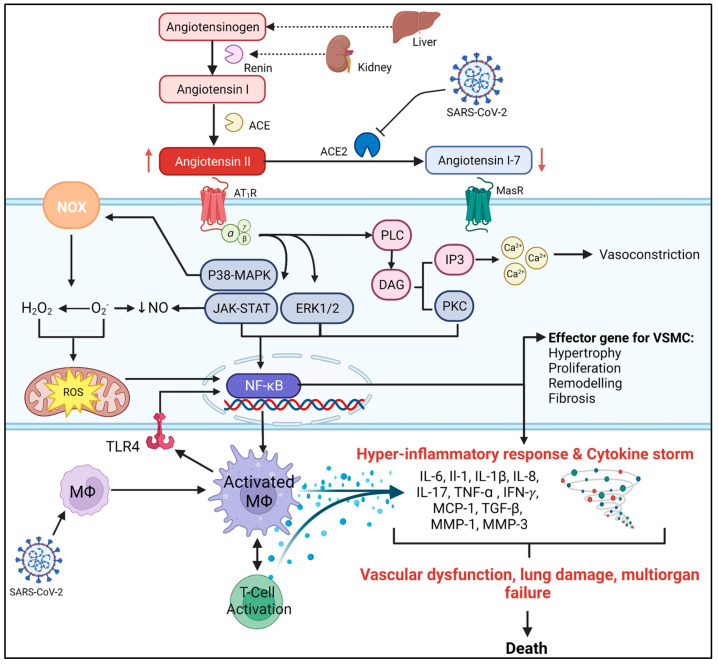
AngII/AT_1_R signaling pathways contribute to the immunopathological features of COVID-19. AngII is produced by the conversion of angiotensinogen from the liver into AngI by the release of renin from the kidney. AngI is metabolized by ACE into AngII, which can bind to the G-protein-coupled AT_1_R. The interaction of ACE2 with SARS-CoV-2 results in a loss of its functional expression and thus decreases its ability to metabolize AngII into the protective Ang (1–7). This interaction is marked by the excessive activation of AT_1_R signaling to promote tissue-dependent signal transduction pathways. In VSMCs, AngII/AT_1_R signaling results in the downstream formation of PLC, leading to the generation of DAG and activation of PKC and IP3. IP3 is then free to release Ca^2+^ into the cell cytosol, where it initiates vasoconstriction. Additionally, AngII/AT_1_R signaling can activate various protein kinase signaling pathways (p38-MAPK, JAK-STAT, ERK, and PKC), which are all involved in the pathogenic effects of AngII. This includes AngII-dependent oxidative stress through activation of NOX, resulting in the production of ROS, O_2_^−^ and H_2_O_2_ that can damage cellular components, upregulate NF-κB to induce the production of pro-inflammatory cytokines, and disrupt endothelial function by decreasing the bioavailability of NO. AngII-dependent upregulation of NF-κB through kinase pathways can also directly induce immune cell activation, proliferation, and differentiation, as well as mediate the production of pro-inflammatory, pro-fibrotic, and pro-hypertrophic mediators. The NF-κB pathway is also involved in the pathogenesis of COVID-19 by inducing the production of pro-inflammatory mediators involved in the progression of the cytokine storm. The inflammatory response to SARS-CoV-2 is established via macrophage activation and TLR4 signaling to create an imbalance in the immune response and promote the excessive amplification of cytokine production that could be combined with the AngII/AT_1_R signaling cascade to result in the progression of endothelial dysfunction, lung injury, and multi-organ failure in severe and fatal states of COVID-19. Abbreviations: ACE, angiotensin converting enzyme; ACE2, angiotensin converting enzyme 2; AT_1_R, angiotensin type I receptor; AngI, angiotensin I; AngII, angiotensin II; Ang (1–7), angiotensin (1–7); Ca^2+^, calcium ions; COVID-19, coronavirus disease 2019; DAG, diacylglycerol; ERK, extracellular signal-regulated kinase; H_2_O_2_, hydrogen peroxide; IP3, inositol-1,4,5-triphosphate; IFN-γ, interferon-gamma; IL, interleukin; JAK-STAT, Janus kinases-STAT; MΦ, macrophage; MasR, Mas1 oncogene receptor; p38-MAPK, mitogen-activated protein kinases; MCP-1, monocyte chemoattractant protein-1; MMP, matrix metalloproteinase; NF-κB, nuclear factor kappa B; NO, nitric oxide; NOX, nicotinamide adenine dinucleotide phosphate oxidase; O_2_^−^, superoxide ion; PKC, protein kinase C; PLC, phospholipid-C; ROS, reactive oxygen species; SARS-CoV-2, severe acute respiratory syndrome coronavirus-2; T-cell, T-lymphocyte; TLR4, toll-like receptor 4; TGF-β, growth factor-beta 1; TNF-α: tumour necrosis factor-alpha; VSMCs: vascular smooth muscle cells. Figure created with biorender.com.

**Figure 2 biomolecules-13-00787-f002:**
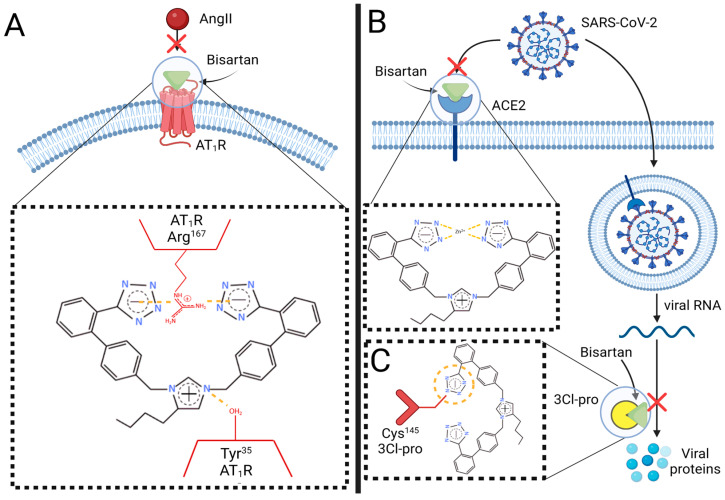
Multifunctional targeting ability of bisartans against SARS-CoV-2 infection and COVID-19 progression. (**A**) Schematic representation of the interaction between the two anionic tetrazole groups of bisartan with AT_1_R via salt-bridge bonds (yellow dashed lines) between Arg^167^ and Tyr^35^, respectively, to provide overwhelming inhibition of receptor activation by AngII. (**B**) Binding of the anionic tetrazole groups of bisartan with the Zn^2+^ active site in ACE2 to prevent viral S-protein interaction and thus disrupt viral entry via ACE2. (**C**) Docking of one tetrazole group of bisartan to residual Cys^145^ in the proteo-catalytic site of 3CL-pro to disrupt its catalytic ability to mediate viral replication. Abbreviations: ACE2, angiotensin converting enzyme-2; AT_1_R, angiotensin type 1 receptor; AngII, angiotensin II; COVID-19, coronavirus 2019; SARS-CoV-2, severe acute respiratory syndrome coronavirus 2; S-protein, spike protein; 3Cl-pro, 3-chymotrypsin-like protease. Figure created with biorender.com.

**Table 1 biomolecules-13-00787-t001:** Summary of clinical studies associated with the protective effects of ARBs in relation to COVID-19.

Subject	ARB/Dose	Outcome	Ref
103 patients with hypertension and COVID-19 receiving ARB therapy	NA	Pre-administered ARBs reduced the need for mechanical ventilation, intensive care admission, and death.	[149]
157 patients diagnosed with COVID-19 and hypertension on ARBs	NA	ARB group is associated with decreased mortality.	[150]
201 hospitalized patients diagnosed with COVID019	NA	Pre-administered ARB had a lower mortality rate compared to other antihypertensive medications.	[151]
636 COVID-19 patients,1 of which 22 receiving ARBs	NA	Discontinuing ARB therapy during COVID-19 infection resulted in greater mortality, ventilation, and increased risk of acute kidney injury.	[152]
19 586 patients with COVID-19	NA	ARBs associated with reduced risk of COVID-19 in patients with hypertension.	[163]
566 hypertensive patients with COVID-19, 147 on ARBs.	NA	ARB therapy resulted in a lower risk of mortality in patients with a high prognostic factor, low oxygen saturation, and high lymphocyte count than other RAS inhibitors.	[164]
63,969 hospitalized participants with COVID-19	Telmisartan	Telmisartan showed a reduction in mortality risks greater than standard care.	[162]
52 COVID-19-diagnosed patients not receiving antihypertensive medication	Telmisartan 160 mg/day, 14 days	Telmisartan reduced morbidity and mortality of COVID-19 patients through anti-inflammatory effects.	[158]
1946 patients with COVID019 and cardiovascular comorbidities of which 493 on ARB	NA	ARBs reduced mortality, leukocyte count, inflammatory markers, and IL-6. No changes in ACE2 expression were observed.	[165]
178 patients with COVID-19 of which 133 used ARBs.	NA	ARBs reduced mortality in patients hospitalized with COVID-19.	[166]

## Data Availability

Not applicable.

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
