# Peer review of "Role of Angiotensin II in Cardiovascular Diseases: Introducing Bisartans as a Novel Therapy for Coronavirus 2019"

_biomolecules, 2023, doi:10.3390/biom13050787_

Round 1
Reviewer 1 Report
This review is a compendium of the relationship between cardiovascular diseases and covid-19 pathology. The authors try to explore the link between the common ARBs and its use as potential drugs for SARS-COV2, also the repurposing of sartans and a new class named bisartans is explored. The manuscript is robust and fit in to the scopes of the Journal. Therefore, this referee believes that it deserves to be published in
Biomolecules, pending the following major points:
· The novel class of ARBs, named “bisartans” are mentioned until the conclusion section. This reviewer suggests to include a special section to describe them before the conclusion.
· Include a section of the most common toxicological effects of angiotensin II receptor antagonist.
· Authors claim that Bisartans were shown to undergo a more stable interaction with the cystine145 residue of 3ClPRO than the nitrile group of Nirmatrelvir, but the last interaction is more stable due to the formation of covalent bond. Please explain how the interaction with tetrazole and cysteine ​​could be more favorable?
· Some parts are very similar to those published in Comput Struct Biotechnol J. 2022;20:2091-2111. doi: 10.1016/j.csbj.2022.04.010
Author Response
Reviewer 1#
This review is a compendium of the relationship between cardiovascular diseases and covid-19 pathology. The authors try to explore the link between the common ARBs and its use as potential drugs for SARS-COV2, also the repurposing of sartans and a new class named bisartans is explored. The manuscript is robust and fits into the scopes of the Journal. Therefore, this referee believes that it deserves to be published in Biomolecules, pending the following major points:
Comments:
Comment 1: The novel class of ARBs, named “bisartans” are mentioned until the conclusion section. This reviewer suggests including a special section to describe them before the conclusion.
Thank you for the review of the manuscript and this comment. We agree that an introduction paragraph to bisartans is required before the conclusion. We have now added section 5. Bisartans, a new generation of ARBs against SARS-CoV-2, which was originally the conclusion section. We have also added 6. Conclusion, which reads as followed:
Line 518-529: Herein, this literature review highlights the profound dysregulation of the RAS in the pathogenesis of COVID-19. We present current evidence that CVDs may exacerbate COVID-19 severity via AngII-dependent mechanisms and viral targeting of ACE2, resulting in a dysfunction and reduced expression of the protective counter-regulatory axis (ACE2/Ang(1-7)/MasR) and an overexpression of the deleterious classical axis (AngII/AT1R) of the RAS, characterised by vasoconstriction, vascular remodeling, oxidative stress, and hyper-inflammation. Due to this association, we show that ARBs may be an effective therapeutic treatment of COVID-19, for which we present “bisartans”, a novel class of AT1R inhibitors that also exhibit multifaceted antiviral abilities against SARS-CoV-2 infection. Therefore, bisartans may be a promising therapeutic treatment for hypertension, CVDs, and COVID-19.
Comment 2: Include a section of the most common toxicological effects of angiotensin II receptor antagonist.
Thank you for bringing this to our attention. To address this, we have now added the following to discuss the common toxicological effects of ARBs:
Line 393-401: ARBs are known to have a low incidence of side effects and contain an excellent tolerability profile comparable to placebo [128,129]. Randomized control trials have demonstrated that the most common adverse effects of ARB treatment include headaches, dizziness, and fatigue [130,131,132]. In rare cases, major adverse effects of ARBs are hemodynamic, resulting in an increased risk of hypotension [133,134]. Additionally, in patients with renal disease, ARBs may increase the risk of hyperkalemia due to the direct effect of AT1R inhibition on renal electrolyte balance [135]. It is important to recognize that these effects are primarily pronounced in patients taking a combination of ARBs and other blood pressure lowering medication.
Comment 3: Authors claim that Bisartans were shown to undergo a more stable interaction with the cystine145 residue of 3ClPRO than the nitrile group of Nirmatrelvir, but the last interaction is more stable due to the formation of covalent bond. Please explain how the interaction with tetrazole and cysteine ​​could be more favorable?
Thank you for this comment. We have addressed the clarity on how the interaction of bisartan tetrazoles with the cystine residual of 3CL-pro could be more favourable than the nitril group of nirmatrelvir by producing a greater disruption of the protease mechanism required for viral replication as follows:
Line 491-499: Additionally, bisartans where identified to be more effective than Nirmatrelvir, the active compound of Pfizer’s antiviral COVID-19 drug in disrupting the CRS of 3-chymotrypsin-like protease (3Cl-pro), a protease required for SARS-CoV-2 replication which we have identified as a key target for drug development against viral infection [12,175]. The ionic tetrazole groups of bisartans were found to have a more stable interaction with the Cys145 residual of 3Cl-pro than the nitril pharmacophoric group of Nirmatrelvir, resulting in greater inhibition of the CRS protease mechanism [175] (Figure 2). As a result, bisartans may be more effective than Nirmatrelvir in preventing the synthesis of essential SARS-COV-2 proteins required for viral replication in humans.
Comment 4# Some parts are very similar to those published in Comput Struct Biotechnol J. 2022;20:2091-2111. doi: 10.1016/j.csbj.2022.04.010
Thank you for bringing this to our attention. Section 5 “Bisartans, a new generation of ARBs against SARS-CoV-2” contains evidence from our previous work published in “Comput Struct Biotechnol J. 2022;20:2091-2111. doi: 10.1016/j.csbj.2022.04.010” on the binding characteristics of bisartans to targets of SARS-COV-2. To address similarities between this review and our previous study, we have re-worded section 5.
Reviewer 2 Report
I carefully read the article sent. the presented ideas are very pertinent from the point of view of the Covid-19 infection as a vascular cardio- vascular disease with a pulmonary response through the prism of ACE intervention.
The authors present in detail the mechanisms of RAS production, such as the involvement of AngII in cardiovascular disease and oxidative stress and inflammation. The existing therapeutic options are also presented, as well as the proposal of a new therapy (bisartans)
I believe that the authors omitted the presentation of the relationship between ACE and myocarditis and sudden deaths.
ACE2 is used by SARS-CoV-2 to initiate the COVID-19 infection, which may downregulate ACE2, leading to additional toxic overaccumulation of angiotensin II that induces acute respiratory distress syndrome and fulminant myocarditis.
Author Response
Reviewer #2:
I carefully read the article sent. the presented ideas are very pertinent from the point of view of the Covid-19 infection as a vascular cardio- vascular disease with a pulmonary response through the prism of ACE intervention. The authors present in detail the mechanisms of RAS production, such as the involvement of AngII in cardiovascular disease and oxidative stress and inflammation. The existing therapeutic options are also presented, as well as the proposal of a new therapy (bisartans)
Comments 1: I believe that the authors omitted the presentation of the relationship between ACE and myocarditis and sudden deaths. ACE2 is used by SARS-CoV-2 to initiate the COVID-19 infection, which may downregulate ACE2, leading to additional toxic overaccumulation of angiotensin II that induces acute respiratory distress syndrome and fulminant myocarditis.
Thank you for bringing this to our attention. We have since added the following to address this comment:
Line 281-290: In patients with acute respiratory distress syndrome and admitted to the intensive care unit, plasma ratio of Ang(1-7) was observed to be higher in survivors than non-survivors suggesting that a grater ACE2 expression may be a protective factor against severe lung injury due to the ability of ACE2 to degrade AngII into the protective peptide Ang(1-7) [88]. ACE2 deficiency in mice with heart failure due to myocardial infarction, an increase myocardial infarct size was associated with an increase in ERK1/2/JNK molecular signalling pathways to increase the expression of IL-6, MCP-1, and MMP-9. Furthermore, overexpression of ACE2, Ang(1-7) and ALM receptor MrgD has been shown to improve cardiac remodelling during myocardial infarction [89,90,91,92].
Line 297-300: Inflammation of the heart myocardium (myocarditis) is recognized as a high-risk mortality factor for heart failure in COVID-19 due to the augmented increase in inflammatory mediators that occurs in the cytokine storm [11].
Line 346-349: Injection of the SARS-COV-2 S-protein into wild-type mice resulted in a decreased expression of ACE2 in the lungs, which was associated with significant increases in AngII, respiratory inflammation, vascular permeability, and decreased oxygen saturation [116].
Line 356-361: Elevated levels of IL-6 have been identified as a major factor in the exacerbation of COVID-19 inflammatory response, contributing to vascular inflammation, acute respiratory distress syndrome development, and mechanical ventilation [119]. Overstimulation of AngII/AT1R-induced signaling is observed to promote endothelial dysfunction, contributing the development of ARDS as characterized by the onset of severe hypoxia, inflammatory cell accumulation and pulmonary oedema [11].
Reviewer 3 Report
The review is well-written, detailed and informative. I am happy to recommend publication. My only slight concern is that the Conclusions section is used to describe published results obtained by the authors. It might be more useful to make most of this section part of the main narrative and have a shorter concluding section afterwards.
Author Response
Reviewer #3:
The review is well-written, detailed, and informative. I am happy to recommend publication.
Comment 1:My only slight concern is that the Conclusions section is used to describe published results obtained by the authors. It might be more useful to make most of this section part of the main narrative and have a shorter concluding section afterwards.
Thank you for your feedback on our manuscript. We have since re-written the conclusion section as followed:
Line 518-529: Herein, this literature review highlights the profound dysregulation of the RAS in the pathogenesis of COVID-19. We present current evidence that CVDs may exacerbate COVID-19 severity via AngII-dependent mechanisms and viral targeting of ACE2, resulting in a dysfunction and reduced expression of the protective counter-regulatory axis (ACE2/Ang(1-7)/MasR) and an overexpression of the deleterious classical axis (AngII/AT1R) of the RAS, characterised by vasoconstriction, vascular remodelling, oxidative stress, and hyper-inflammation. Due to this association, we show that ARBs may be an effective therapeutic treatment of COVID-19, for which we present “bisartans”, a novel class of AT1R inhibitors that also exhibit multifaceted antiviral abilities against SARS-CoV-2 infection. Therefore, bisartans may be a promising therapeutic treatment for hypertension, CVDs, and COVID-19.
Reviewer 4 Report
The manuscript summarizes the detrimental effects of angiotensin 1 receptor activation on cardiovascular disease and SARS-CoV-2 infection.
Strengths:
1. Comprehensive and almost complete, concerning the cardiovascular, renal, pulmonary, immune effects of angiotensin II
2. Ang(1-7) as an important antagonist of angiotensin II, with potent vasoprotectant properties, is highlighted.
Weaknesses:
1. There may be other antagonists to angiotensin II such as angiotensin(2-8) which have been shown to have own receptors and functions, especially in the brain (JW Harding, JW Wright, RL Haberl). Is there any evidence that they - or the aminopeptidases producing them - have a role in cardiovascular disease or SARS-CoV-2 prevention?
Author Response
Reviewer #4:
The manuscript summarizes the detrimental effects of angiotensin 1 receptor activation on cardiovascular disease and SARS-CoV-2 infection. Strengths:
- Comprehensive and almost complete, concerning the cardiovascular, renal, pulmonary, immune effects of angiotensin II
- Ang(1-7) as an important antagonist of angiotensin II, with potent vasoprotectant properties, is highlighted.
Comments 1: There may be other antagonists to angiotensin II such as angiotensin(2-8) which have been shown to have own receptors and functions, especially in the brain (JW Harding, JW Wright, RL Haberl). Is there any evidence that they - or the aminopeptidases producing them - have a role in cardiovascular disease or SARS-CoV-2 prevention?
Thank you for your review of our manuscript. To address your comment on evidence pertaining to other renin angiotensin system peptides and the aminopeptidases producing them, we have included:
Line 100-118: AngII is also metabolised by aspartate decarboxylase to produce angiotensin A (AngA), which can be converted into the recently discovered peptide almandine (ALM) by ACE2 [24]. ALM has been observed to exhibit protective AngII antagonistic effects, similar to Ang(1-7) through activation of Mas-related G-protein coupled receptor D (MrgDR) signalling pathway [25]. Our group has previously highlighted the cardioprotective role of ALM in mitigating endothelial dysfunction and producing antihypertensive effects through MrgDR activation [26]. ALM has also been shown to provide anti-inflammatory [27,28], anti-oxidative [29,30], and pulmonary protective effects [31]. Moreover, AngII can be further processed by aminopeptidase A to form angiotensin (2-8) (Ang III) [32]. AngIII has a similar affinity to the AT1R as AngII [33]. Its major physiological role is in the brain where it acts are a central regulator of blood pressure and vasopressin release [34]. AngIII can be cleaved by alanyl aminopeptidase N to generate angiotensin IV (AngIV), which has been shown to bind to the angiotensin type 4 receptor, leading to mitigation of cardiac hypertrophy [35],fibrosis [35], and inflammation [36], as well as inducing vasoconstriction through nitric oxide release [37,38]. Interestingly, there is evidence that AngIV may act as a weak agonist of the AT1R, resulting in vasoconstriction [39]. The cardioprotective role of ALM and AngIV remain largely elusive and further investigation is required to understand their physiological role in the cardiovascular system.
Line 290-294: Several case reports, along with clinical phase I and II studies have reported that recombinant ACE2 may improve the clinical course of patients with COVID-19 by increasing the degradation of AngII into Ang(1-7), as well a increasing ALM [96,97,98], therefore, downregulating the toxic pathophysiological effects of AngII while upregulating the protective counter-regulator axis.
Reviewer 5 Report
In this manuscript, the authors reviewed the role of angiotensin II in cardiovascular disease and its upregulation in COVID-19. They also demonstrated the potential of a new class of angiotensin receptor blockers, bisartans, that are likely to multifunctionally target at COVID-19 and may be used as a multifunctional drug potentially preventing SARS-CoV-2 infection. The review paper is well composed and clearly written. It contains all the necessary information needed to understand the given topic.
I have only minor comments on Figure 1: please put the names MCP-1 and MMP-1 together without separation when moving to the next line.
Author Response
Reviewer #5:
In this manuscript, the authors reviewed the role of angiotensin II in cardiovascular disease and its upregulation in COVID-19. They also demonstrated the potential of a new class of angiotensin receptor blockers, bisartans, that are likely to multifunctionally target at COVID-19 and may be used as a multifunctional drug potentially preventing SARS-CoV-2 infection. The review paper is well composed and clearly written. It contains all the necessary information needed to understand the given topic.
Comment 1: I have only minor comments on Figure 1: please put the names MCP-1 and MMP-1 together without separation when moving to the next line.
Thank you for your comments on this manuscript, and for bringing the minor amendment to Figure 1 to our attention. We have now put MCP-1 and MMP-1 together and moved them to the next line in order to remove the separation.
Please note that we have proofread the entire manuscript to correct any other spelling mistakes and grammatical errors.
Round 2
Reviewer 1 Report
Authors have improved and corrected all the comments made by the reviewers in the first original submission. Now, the article is suitable for publication.
Author Response
Thanks for your review.